# Correlation of Montmorillonite Sheet Thickness and Flame Retardant Behavior of a Chitosan–Montmorillonite Nanosheet Membrane Assembled on Flexible Polyurethane Foam

**DOI:** 10.3390/polym11020213

**Published:** 2019-01-26

**Authors:** Peng Chen, Yunliang Zhao, Wei Wang, Tingting Zhang, Shaoxian Song

**Affiliations:** 1School of Resources and Environmental Engineering, Wuhan University of Technology, Luoshi Road 122, Wuhan 430070, China; peng141592@163.com (P.C.); whutwangwei@126.com (W.W.); ajting0419@163.com (T.Z.); 2Hubei Key Laboratory of Mineral Resources Processing and Environment, Wuhan University of Technology, Luoshi Road 122, Wuhan 430070, China

**Keywords:** MMTNS, FPU foam, flame retardancy

## Abstract

Polymer–clay membranes constructed via the layer-by-layer (LbL) assembly, with a nanobrick wall structure, are known to exhibit high flame retardancy. In this work, chitosan–montmorillonite nanosheet (CH–MMTNS) membranes with different thickness of MMTNS were constructed to suppress the flammability of flexible polyurethane (FPU) foam. It was found that a thinner MMTNS membrane was more efficient in terms of reducing the flammability of the FPU foam. This was because such MMTNS membrane could deposit cheek by jowl and form a dense CH–MMTNS membrane on the foam surface, thus greatly limiting the translation of heat, oxygen, and volatile gases. In contrast, a thicker MMTNS constructed a fragmentary CH–MMTNS membrane on the coated foam surface, due to its greater gravity and weaker electrostatic attraction of chitosan; thus, the flame retardancy of a thick MMTNS membrane was lower. Moreover, the finding of different deposition behaviors of MMTNS membranes with different thickness may suggest improvements for the application of clay with the LbL assembly technology.

## 1. Introduction

As a complex three-dimensional material, flexible polyurethane (FPU) foam is widely applied in various fields, such as furniture, packaging, automobiles, etc. [1,2], primarily because of its excellent cushioning and physical properties [3]. Unfortunately, FPU foam is highly flammable because of its chemical nature and high air permeability [4], causing a potential fire hazard when it is used. In order to limit its flammability, FPU foam is usually filled with flame retardants, such as organohalogen and phosphorous compounds [5]. However, concerns about the potential negative environmental and health impacts of these flame retardants are growing, which urges us to find safer replacements [6].

As a nanofabrication technique for the preparation of controlled layered structures, layer-by-layer (LbL) assembly can integrate various polymers, nanoparticles, and molecules into a thin film utilizing the driving forces of electrostatic interactions, hydrogen bonds, and van der Waals interactions [7]. This precisely controlled deposition has been extensively applied to build a variety of multifunctional thin films owing to the advantages of simplicity and efficiency [8,9]. Moreover, LbL assembly is also a prevalent method to limit the flammability of inflammables, because an efficient fire hazard suppression system can be formed on the surface of inflammable materials with this method.

Montmorillonite (MMT), a natural layered mineral, has been widely used in many fields, such as drug delivery [10], energy storage [11], and adsorption [12], because of its advantages of low cost, cation-exchange capability, and naturally abundance. In addition, montmorillonite nanosheets (MMTNS), the exfoliated product of MMT, has the distinctive properties of high surface area, natural negative charge, and adiabatic characteristics. Thus, MMTNS is extensively utilized to form thin membranes on the surface of inflammables via LbL assembly to provide protection through a condensed phase mechanism [13]. MMTNS and β-FeOOH nanorods were coated on the surface of FPU foam by Wang et al. [14] via LbL assembly to limit FPU flammability. It was found that the peak heat release rate (PHRR) of the coated foam was sharply reduced, as a result of the formation of a sandwich-like topology on the FPU foam surface. Laufer et al. [15] deposited MMTNS and positively charged chitosan on the surface of FPU foam via the LBL assembly technique. The PHRR of the foam could be reduced by 52% with a 30-bilayer (Chitosan pH 6–MMT) nanocoating.

Although a large number of studies have revealed the excellent performance of MMTNS in reducing the flammability of FPU foam, there are few studies examining the effects of MMTNS thickness on the flammability of FPU foam. However, MMTNS thickness does need to be investigated, because lamellar thickness is not only the most important characteristic index of MMTNS but also the main different feature between MMTNS and MMT.

In this work, in order to investigate the effect of MMTNS thickness on suppressing the flammability of FPU foams, MMTNS with different thicknesses were alternately deposited with chitosan on the surface of FPU foams, and both thermal stability and flammability of the resulting coated foams were tested. Another objective of this work was to investigate the mechanism at the basis of the different performances of MMTNS depending on their thickness in flame retardancy, considering their native micro-morphology, surface potential, and the topography of the FPU foam surface after deposition.

## 2. Materials and Methods

### 2.1. Materials

The flexible polyurethane foam was obtained from Daye Tengfei Sponge Factory, Changzhou, China. Montmorillonite was provided by Ningcheng Tianyu Bentonite Technology Co., Ltd. (Inner Mongolia, Chifeng, China). Polyacrylic acid (PAA) (35%, MW = 100,000 g/mol) was purchased from Sigma-Aldrich. Chitosan was purchased from Sinopharm Chemical Reagent Co. Ltd. (Shanghai, China). Sodium hydroxide, acetic acid (99.5%), nitric acid, and hydrochloric acid (36–38%) were all purchased from Changzheng Chemical Reagent Corp. (Hangzhou, China). The water used in this work was produced by a Millipore Milli-Q Direct 8/16 water purification system with 18.2 MΩ.

### 2.2. Procedures and Methods

#### 2.2.1. Preparation of MMTNS-1 and MMTNS-2

A total of 50 g MMT powder was added to 1000 mL deionized water and stirred for 5 h with a mechanical stirrer at a speed of 1000 rpm. Subsequently, the colloidal suspension was centrifuged at 1000 rpm for 3 min to obtain MMTNS-1.

MMTNS-2 was prepared according to an ultrasonic process rather similar to that introduced in our previous work [16]. The MMT suspension was treated by a Cole Parmer ultrasonic processor (750 W and 20 kHz) with 60% amplitude for 4 min. Finally, the colloidal suspension was centrifuged at 13,000 rpm for 4 min to remove the unexfoliated product, and the homogeneous supernatant was the MMTNS-2 suspension.

#### 2.2.2. Preparation of Chitosan and PAA Solutions

The chitosan solution (0.5 wt %) was prepared by adding chitosan to an acetic acid solution (2.5 wt %); then, the solution was stirred for 3 h, and the pH was adjusted to 3.5 using 1.0 M NaOH. PAA solution was prepared as a 1 wt % solution using deionized water, and then the pH of the PPA solution was adjusted to 2 using 2 M HNO_3_.

#### 2.2.3. LbL Assembly Deposition Process

Firstly, the FPU foams were pretreated by soaking in HNO_3_ (0.1%) for 5 min; the excess acidic solution was then squeezed out to create a positive charge on the surface. Subsequently, the FPU foams were soaked in the PAA solution for 5 min to improve adhesion. After pretreatment, the FPU foams were soaked in sequence in chitosan solution and MMTNS-1 or MMTNS-2 suspensions. Each dip was followed by rinsing with deionized water for 2 min and squeezing to expel the excess liquid from the FPU foams. The procedure is shown in Figure 1. A bilayer was deposited after a rinse-and-dip cycle. After the desired number of bilayers was deposited, the coated FPU foams were dried at 60 °C overnight before testing. In this work, on each FPU foams two bilayers were deposited, and the additions of MMTNS-1 and MMTNS-2 were 4.65% and 2.44%, respectively. The reason for the higher percentage of MMTNS-1, despite the same number of layers, is that the thickness of MMTNS-1 was much greater than that of MMTNS-2.

### 2.3. Characterization

X-ray diffraction (XRD) measurements of MMT particles were conducted with an X-ray diffractometer (D/MAX-RB, Japan); the 2θ of the samples was swept from 5° to 80°. The surface topographies of MMTNS-1 and MMTNS-2 were imaged using a MultiMode 8 AFM (Bruker, Santa Barbara, CA, USA) with Peak Force Tapping mode in air. The morphologies of control and treated flexible PU foams, coated with a gold layer in advance, were observed using scanning electron microscopy (SEM) (JSM-5610LV, Tokyo, Japan). Thermosgravimetric analysis (TGA) of the samples under nitrogen atmosphere was performed by a Simultaneous Thermal Analyzer (STA449F3, Shanghai, China) from 50 to 700 °C at a heating rate of 20 °C /min. The combustion test was performed on a cone calorimeter (China General Nuclear Power State Key Laboratory) using a standardized procedure (ASTM E-1354-07), with 10 × 10 × 2.5 cm^3^ specimens. Each specimen was exposed horizontally to a 35 kW/m^2^ external heat flux. Zeta potential analysis of MMTNS was carried out on a Malvern Zetasizer Nano ZS90 (Malvern, UK) equipped with a rectangular electrophoresis cell at 25 °C.

## 3. Results and Discussion

### 3.1. Characterization of MMTNS

The XRD patterns of MMT, MMTNS-1, and MMTNS-2 are shown in Figure 2. The diffraction peaks of each sample could be readily indexed to the monoclinic crystal system of MMT (JCPDS 13-0135) [17], and no peaks of other materials were observed, which indicated the high purity of the raw material and the exfoliated products. Furthermore, the peak (001) of MMTNS-2 was much broader than that of MMT and MMTNNS-1, which indicated high exfoliation of MMTNS-2.

Typical 2D AFM morphology images (4 × 4 μm, 512 pixels) of MMTNS-1 and MMTNS-2 are shown in Figure 3a,b. It is obvious that the morphology of exfoliated MMT was in the form of few stacked nanosheet with varying thickness. A cross-section analysis along the red line in the AFM images of MMTNS-1 and MMTNS-2 is depicted in Figure 3c,d. It can be seen that the thickness of MMTNS-1 was about 13 nm, whereas the thickness of MMTNS-2 was around 1.5 nm, which is in accordance with the theoretical thickness of monolayer MMT [18], indicating the good preparation of MMTNS. Furthermore, the statistical analysis of lamellar thickness is depicted in Figure 3e,f, showing the thicknesses of MMTNS-1 and MMTNS-2 centered around 15–17 nm and 1.5–3 nm, respectively. These results indicated that the thickness of MMTNS-1 was obviously greater than that of MMTNS-2.

### 3.2. Characterization of Thermal Stability and Flame Retardancy

In order to evaluate the thermal degradation behavior of control and coated FPU foams, TGA was applied in this work. Figure 4 presents the TGA and differential thermal gravity (DTG) curves of control and coated FPU foams under N_2_ atmosphere. It is obvious that two typical thermal degradation stages were observed for all samples. The first thermal degradation stage occurred in the range of 215–320 °C with 29% mass loss, due to depolymerization of urethane and the bisubstituted urea groups [19]. Owing to the pyrolysis of the remaining polyether chain [20], the second thermal degradation stage occurred with about 68% mass loss. Furthermore, it can be seen that the protection of the FPU foam by MMTNS-1 primarily occurred in the first stage, whereas, the protection of the FPU foam by MMTNS-2 mainly occurred in the second stage. This was due to the fact that the T_10%_ of the MMTNS-1-coated foam was slightly higher than that of the MMTNS-2-coated foam, whereas the T_50%_ of the MMTNS-2-coated foam was significantly higher than that of the MMTNS-1-coated foam. The residue char is an important index for flame retardancy [21]. In this work, the residue char was obtained by subtracting the addition (MMTNS) from the residue mass. The residue char of control, MMTNS-1-, and MMTNS-2-coated foams after the TGA test were 0.988%, 1.59%, and 2.47% (Table 1), respectively. It is obvious that the residue chars of both MMTNS-1- and MMTNS-2-coated foam were quite higher than that of the control foam, whereas the residue char of the MMTNS-2-coated foam was about 1.55 times that of the MMTNS-1-coated foam. These results revealed that both MMTNS-1 and MMTNS-2 could postpone the degradation of the FPU foam, but MMTNS-2 was more efficient.

To investigate the flammability of the control and coated FPU foams, the foams were initially tested by direct exposure to a flame with a hand-held butane torch for 10 s. The control foam ignited and started to melt drip immediately upon exposure to the flame and completely consumed after 103 s (Figure 5a). No melt dripping was observed for the both MMTNS-1- and MMTNS-2-coated foams, and the flame was extinguished after it traveled across the foam surface (Figure 5b). Chars were formed over the coated foams with a complete retention of the original shape, as shown in Figure 5c. Moreover, it can be seen from Figure 5d that residue foam was observed underneath the char. Figure 5e illustrates the residue percentage of the control and coated FPU foams. It is obvious that the control FPU foam was completely consumed without any residue after exposure to the flame. Both MMTNS-1- and MMTNS-2-coated foams presented a residue foam. The MMTNS-2-coated foam had a residue percentage of 64%, bigger than that of the MMTNS-1-coated foam. This result indicated that the flammability of the FPU foam could be significantly decreased by the deposition of both MMTNS-1 and MMTNS-2, and MMTNS-2 was more effective than MMTNS-1.

A cone calorimeter, simulating a developing fire scenario, was employed to more quantitatively investigate the improvement of flame retardancy for the FPU foam by MMTNS-1 and MMTNS-2 [22]. The heat release rate (HRR) curves of control and coated FPU foams during the cone calorimeter test are illustrated in Figure 6. All foam curves consisted of two peaks associated with the combustion of polyisocyanate and polyol [23]. The two peaks of the coated foams were both much lower than those of the control foam, indicating that two layers of both MMTNS-1 and MMTNS-2 could limit the flammability of the PFU foam. Furthermore, both peaks of the MMTNS-1-coated foam were higher than those of the MMTNS-2-coated foam, revealing that MMTNS-2 was more efficient than MMTNS-1 in suppressing the flammability of the FPU foam. Similar results could also be obtained from the total heat release (THR) curves, because the THR decrement between control and MMTNS-2-coated foam was twice the THR decrement between control and MMTNS-1-coated foam (the table inserted in Figure 6b).

### 3.3. Mechanism by Which MMTNS Thickness Affects the Flame Retardancy of CH–MMTNS membranes

It could be confirmed that MMTNS-2 was more efficient than MMTNS-1 in reducing the flammability of the FPU foam via LbL assembly. However, the result was unexpected, because the flammability of the MMTNS-1-coated foam with more MMT was higher than that of the MMTNS-2-coated foam. In order to explore the mechanism of this difference, SEM was employed in this study to observe the surface morphology of the control and coated FPU foams. SEM images of control foam, MMTNS-1-, and MMTNS-2-coated foams are shown in Figure 7. Compared with the control foam, the surface of the coated foams became very rough, indicating that both MMTNS-1 and MMTNS-2 were successfully deposited on the surface of the FPU foam. Yet, some surface morphology differences between the foams coated by MMTNS-1 and MMTNS-2 under identical assembly condition were observed. It can be seen from Figure 7e that MMTNS-2 uniformly deposited on the FPU foam surface and formed a dense CH–MMTNS membrane, thus protecting the foam from damage when it was exposed to fire. Although most of the FPU foam surface was coated by MMTNS-1 (see Figure 7f), no dense membrane was formed, because some defects formed on the CH–MMTNS membrane in the process of LbL assembly.

The surface morphology of MMTNS-1- and MMTNS-2-constructed membranes indeed presented differences, which might influence the flammability of the FPU foam. A schematic diagram of the flame retardancy behavior of CH–MMTNS membranes constructed with MMTNS at different thickness is illustrated in Figure 8. As it is well known, the flame retardant clay acts as a heat dispersant and is also good at reinforcing char because of its generally high thermal stability [24,25]. The conduction of heat and permeation of oxygen can be reduced in the dense membrane formed on the surface of the MMTNS-2-coated FPU foam, delaying the pyrolysis of the foam, thus limiting the flammability of the FPU foam. Simultaneously, the flame retardancy of the foam can also be improved by the formed char by abating the transfer of heat, oxygen, and volatile gases [14]. In contrast, the blocking effect on the conduction of heat and permeation of oxygen was reduced in the MMTNS-1-coated FPU foam, owing to the formation of defects on the formed membrane in the assembly process, which decreased the flame retardancy of the CH–MMTNS membrane.

Our results suggest that the different deposition morphologies of different MMTNS influence the flame retardancy of a CH–MMTNS membrane. But why was the surface morphology of the MMTNS-1- and MMTNS-2-constructed membranes different? As described in Section 2, the LbL assembly process was accomplished by alternatively depositing MMTNS-1 or MMTNS-2 with chitosan on the FPU foam surface through electrostatic interactions. Therefore, the surface electricity of MMTNS-1 and MMTNS-2 is very important for the LbL assembly. In order to investigate the deposition behavior of MMTNS-1 and MMTNS-2 with chitosan on the FPU foam surface, zeta potentials of both MMTNS-1 and MMTNS-2 were measured under the same concentration at pH 9. The zeta potential distributions of MMTNS-1 and MMTNS-2 are illustrated in Figure 9. It can be seen that the zeta potential peak of MMTNS-2 locates at -42 mV, whereas the peak of MMTNS-1 locates at −25 mV. This result indicates that the electrostatic interaction between MMTNS-2 particles was much greater than the electrostatic interaction between MMTNS-1 particles; thus, MMTNS-2 has higher dispersion stability in aqueous solution [26]. Furthermore, it could also be inferred from the zeta potential results that the electrostatic attraction between the chitosan chain and MMTNS-2 is greater than the electrostatic attraction between the chitosan chain and MMTNS-1.

According to the analytic results of the zeta potential and AFM measurements, a deposition behavior of MMTNS-1 and MMTNS-2 is proposed. A schematic diagram of the deposition behavior of MMTNS-1 and MMTNS-2 is illustrated in Figure 10. When the chitosan-coated foams soak into the MMTNS-1 and MMTNS-2 suspensions, both MMTNS-1 and MMTNS-2 particles, respectively, deposit cheek by jowl on the foam surface through electrostatic attraction. However, in the following step of rinsing, the relatively thicker MMTNS-1 particles will drop out from the foam surface, because of the weaker electrostatic attraction and higher gravity, thus producing a fragmentary CH–MMTNS membrane on the foam surface. In contrast, MMTNS-2 will keep depositing on the foam surface cheek by jowl, because of the strong electrostatic attraction and lower gravity, thus forming a dense CH–MMTNS membrane on the foam surface.

## 4. Conclusions

The thickness of MMTNS-1 was much greater than that of MMTNS-2, whereas the surface potential of MMTNS-2 was much more negative than that of MMTNS-1. As a result of these properties, the thick MMTNS could not form a dense CH–MMTNS membrane on the FPU foam surface, which was observed with the thin MMTNS. This was due to the fact that the relatively thicker MMTNS dropped out during the process of rinsing because of its higher gravity and weaker electrostatic attraction with chitosan, thus forming a fragmentary CH–MMTNS membrane on the surface of the FPU foam. Therefore, the reduction of the translation of heat, oxygen, and volatile gases by the thick MMTNS- coated foam was more limited than that observed for the foam coated by the thin MMTNS. Finally, although both the thick and the thin MMTNS could significantly improve the flame retardancy of the FPU foam via LbL assembly, the thin MTNS was more efficient. Furthermore, the finding about the different deposition behaviors of MMTNS with different thicknesses may help improve the application of clay with the LbL assembly technology.

## Figures and Tables

**Figure 1 polymers-11-00213-f001:**
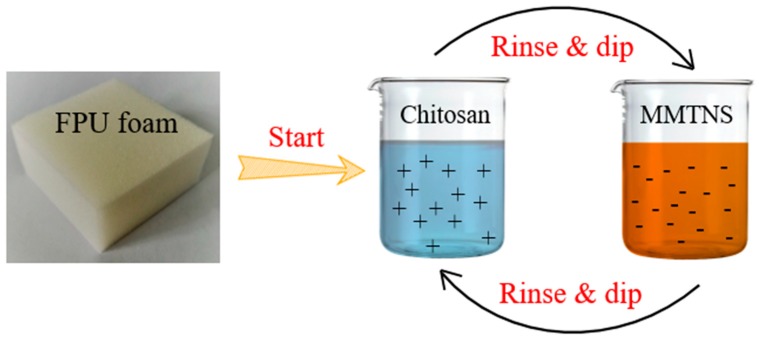
Schematic diagram of layer-by-layer (LbL) assembly on flexible polyurethane (FPU) foam using chitosan and montmorillonite nanosheets (MMTNS).

**Figure 2 polymers-11-00213-f002:**
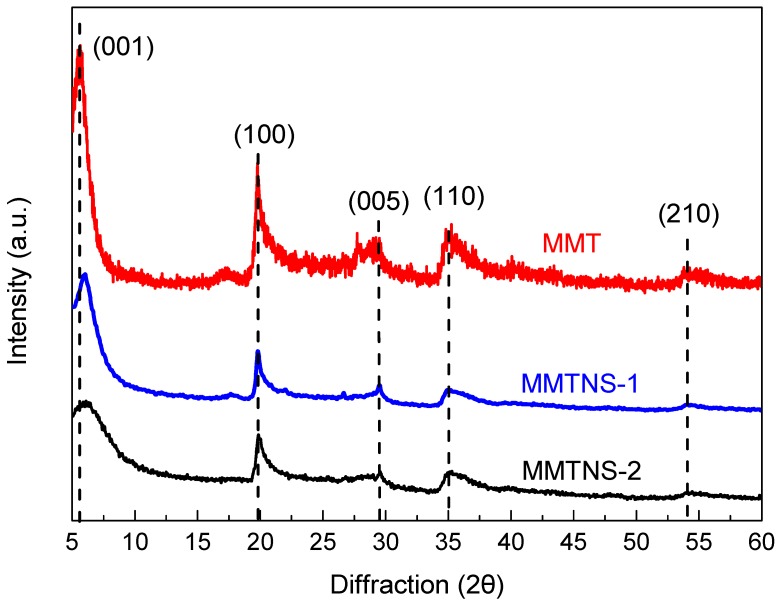
XRD patterns of montmorillonite (MMT), MMTNS-1, and MMTNS-2.

**Figure 3 polymers-11-00213-f003:**
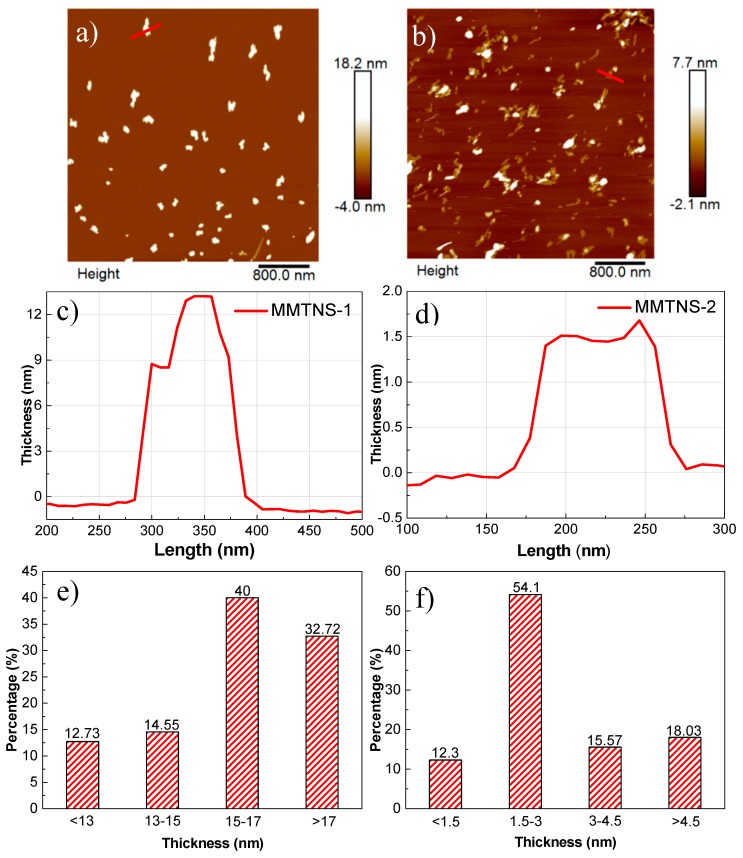
Typical 2D AFM morphology images (4 × 4 μm, 512 pixels) of MMTNS-1 (**a**) and MMTNS-2 (**b**); topographic profile along the red lines in the corresponding images of MMTNS-1 (**c**) and MMTNS-2 (**d**); thickness distribution of MMTNS-1 (**e**) and MMTNS-2 (**f**).

**Figure 4 polymers-11-00213-f004:**
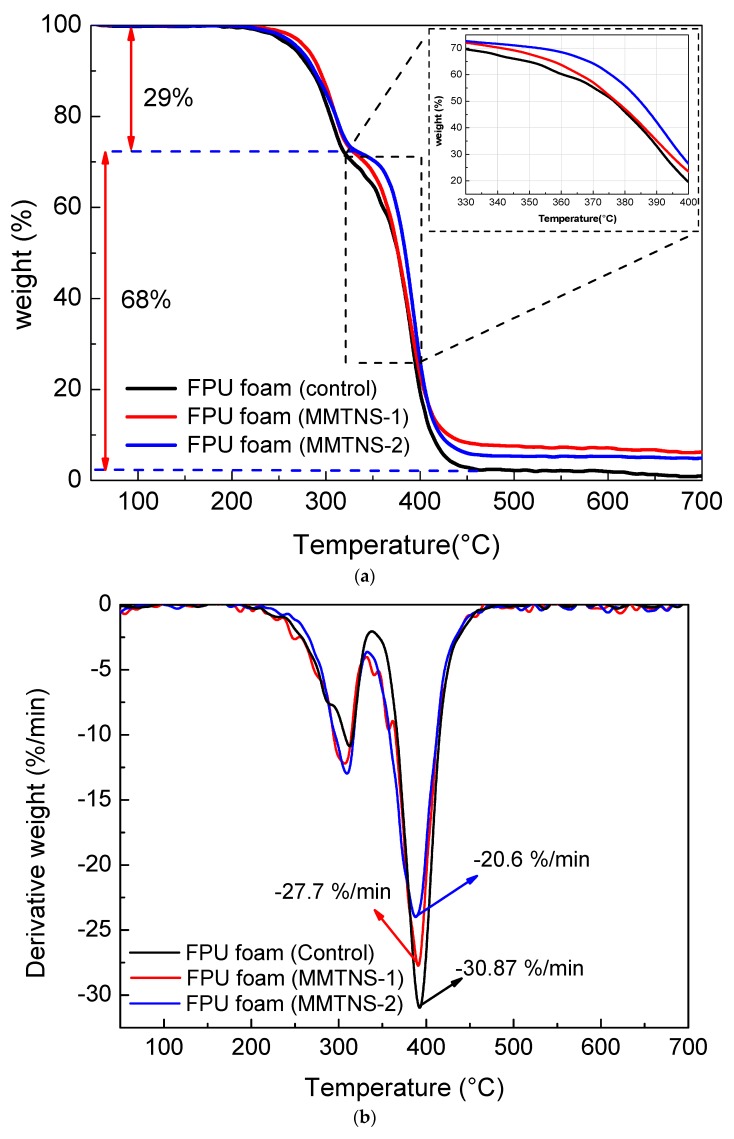
TGA (**a**) and DTG (**b**) curves of control and coated FPU foams under N_2_ atmosphere.

**Figure 5 polymers-11-00213-f005:**
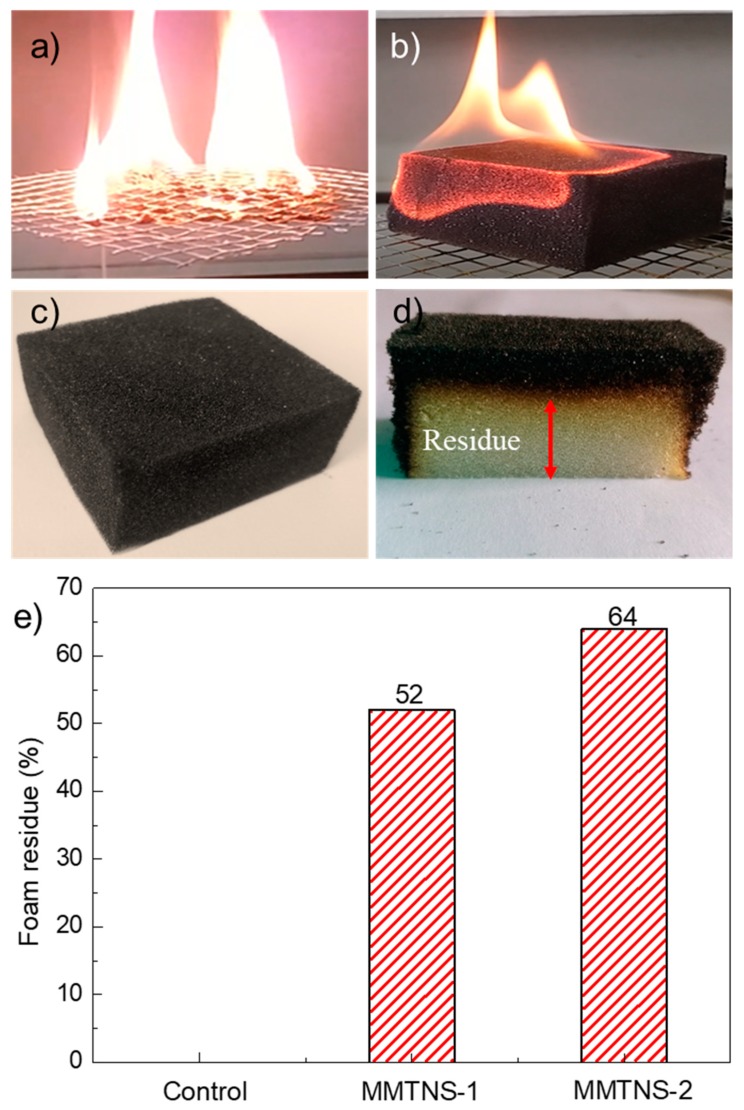
Combustion of control foam (**a**); coated foam (**b**); digital micrographs of charred foam (**c**) and vertically cut foam after butane torch exposure (**d**); residue of FPU foams after direct exposure to the flame (**e**).

**Figure 6 polymers-11-00213-f006:**
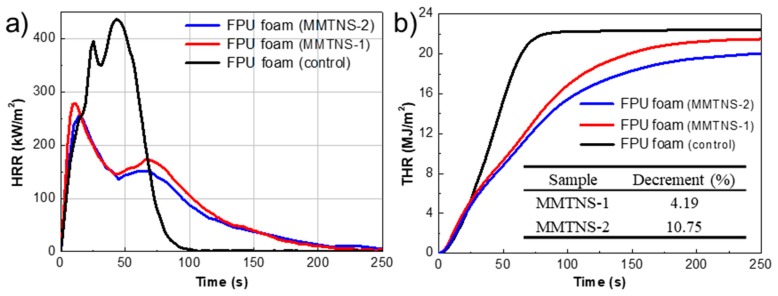
(**a**) Heat release rate (HRR) curves and (**b**) total heat release (THR) curves of control and coated FPU foams during the cone calorimetry test.

**Figure 7 polymers-11-00213-f007:**
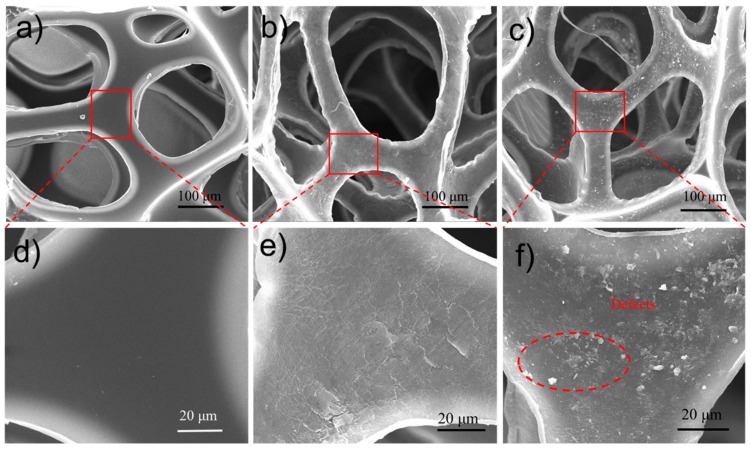
SEM images of control FPU foam (**a**,**d**); MMTNS-2- (**b**,**e**); and MMTNS-1- (**c**,**f**) coated foams t different magnification.

**Figure 8 polymers-11-00213-f008:**
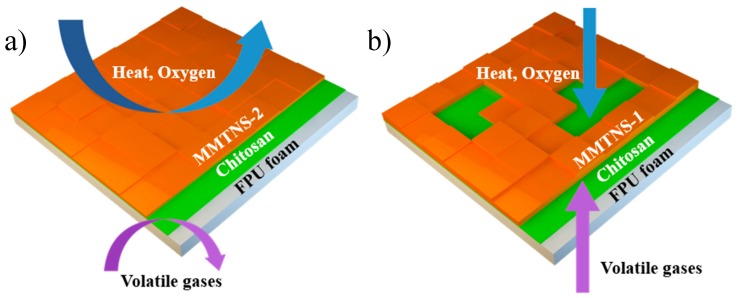
Schematic diagram of the flame-retardant behavior of the CH–MMTNS membranes constructed by MMTNS-2 (**a**) and MMTNS-1 (**b**).

**Figure 9 polymers-11-00213-f009:**
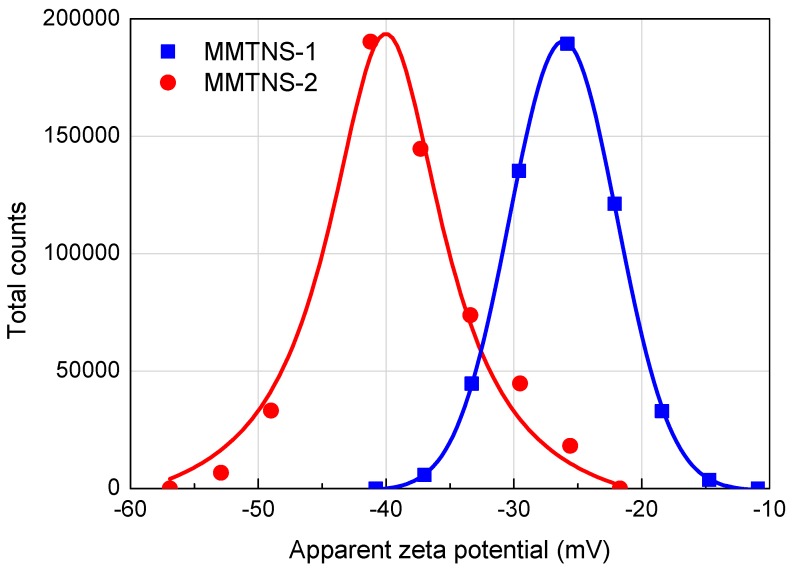
Zeta potential distribution of MMTNS-1 and MMTNS-2 at pH 9.

**Figure 10 polymers-11-00213-f010:**
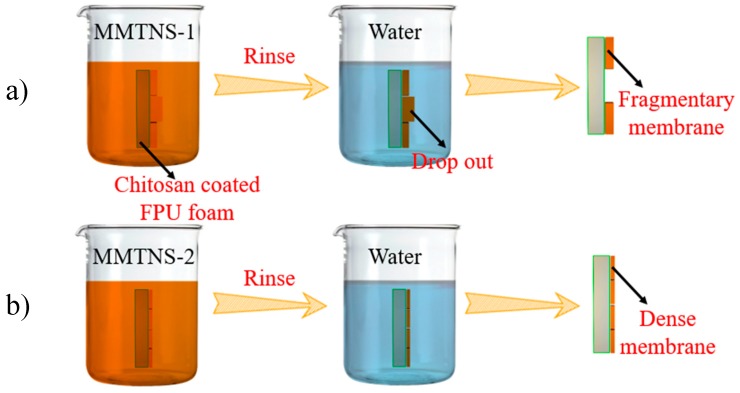
Schematic diagram of the deposition behavior of MMTNS-1 (**a**) and MMTNS-2 (**b**).

**Table 1 polymers-11-00213-t001:** TGA data of control and coated FPU foams.

Sample	Addition (%)	T_5%_ (°C)	T_10%_ (°C)	T_50%_ (°C)	Residue Mass (%)	Residue Char (%)
Control	0	265.7	284.6	376.4	0.988	0.988
MMTNS-1	1.59	279	290.3	376.7	6.24	1.59
MMTNS-2	2.47	277.5	287.6	384.2	4.91	2.47

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
