# Peer review of "Correlation of Montmorillonite Sheet Thickness and Flame Retardant Behavior of a Chitosan–Montmorillonite Nanosheet Membrane Assembled on Flexible Polyurethane Foam"

_polymers, 2019, doi:10.3390/polym11020213_

Round 1

Reviewer 1 Report

In this work, Author prepared the chitosan-montmorillonite nanosheets membranes and modified of flexible polyurethane foam. They analyzed morphology, thermal stability and flame retardancy of “new materials”. The obtained results are very interesting from a scientific and industrial application point of view. In my opinion reviewed paper is suitable for publication in Polymers, however, before acceptance, some minor revision should be performed. My main comments are pointed below:

1.  Highlights should be changed

2. In the Introduction authors should write what it new in this research

3. The author shows the XRD of the MMT – why? (it does not contribute anything to the manuscript). My suggestion: Author should show the XRD of MMT, MMTNS-1, and MMTNS-2 and give some discussions.

4. The thermal stability parameters should be added to this manuscript (T5, T10, and DTG curves.

5. Fig 4. The author shows the TG curses and Residue mass (%) – in table 1, we can see the some – why ?. My suggestion: remove residual mass form the Fig 4.

6. To better thermal analysis I suggest give DTG curves.

Author Response

In this work, Author prepared the chitosan-montmorillonite nanosheets membranes and modified of flexible polyurethane foam. They analyzed morphology, thermal stability and flame retardancy of “new materials”. The obtained results are very interesting from a scientific and industrial application point of view. In my opinion reviewed paper is suitable for publication in Polymers, however, before acceptance, some minor revision should be performed. My main comments are pointed below:

(1) Highlights should be changed

Response: Thank you for your recommendation, and the highlights have changed in the manuscript.

(2) In the Introduction authors should write what it new in this research

Response: Thank you for your suggestion, and the introduction of this manuscript has been polished to emphasize our innovation.

(3) The author shows the XRD of the MMT – why? (it does not contribute anything to the manuscript). My suggestion: Author should show the XRD of MMT, MMTNS-1, and MMTNS-2 and give some discussions.

Response:  XRD patterns of MMT, MMTNS-1 and MMTNS-2 has inserted in Fig. 2, and discussed in chapter 3.1.

(4) The thermal stability parameters should be added to this manuscript (T5, T10, and DTG curves.

Response: The data of T5% and T10% has added in Table 1, and DTG curves have added in Fig. 4.

(5) Fig 4. The author shows the TG curses and Residue mass (%) – in table 1, we can see the some – why ? My suggestion: remove residual mass form the Fig 4.

Response: We agreed with your suggestion, and the residual mass has removed from Fig. 4.

(6) To better thermal analysis I suggest give DTG curves.

Response: We appreciate this suggestion, and DTG curves have added in Fig. 4.

Reviewer 2 Report

The content of this study is interesting and the results are useful for the researchers in the related field. However, there are some problems to be clarified. Therefore, I suggest a major revision of this paper.

1. Even though it seems that the thickness of MMTNS is critical to suppress the flammability of FPU foam, the lateral dimension of MMTNS can be critical to suppress the flammability of FPU foam. Because lateral dimension of MMTNS-2 can be smaller than lateral dimension of MMTNS-1, this issue should be clarified.

2. Please provide the reason that electrostatic interaction between MMTNS-2 particles is much bigger than the electrostatic interaction between MMTNS-1 particles.

3. In p8, "However, the result is unexpected, because the flammability of MMTNS-2 coated foam with a lower addition is higher than that of MMTNS-1 coated foam." This sentence can be wrong.

Author Response

The content of this study is interesting and the results are useful for the researchers in the related field. However, there are some problems to be clarified. Therefore, I suggest a major revision of this paper.

(1) Even though it seems that the thickness of MMTNS is critical to suppress the flammability of FPU foam, the lateral dimension of MMTNS can be critical to suppress the flammability of FPU foam. Because lateral dimension of MMTNS-2 can be smaller than lateral dimension of MMTNS-1, this issue should be clarified.

Response: Thank you for your suggestion, the lateral dimension of MMTNS may have influence on the flammability of FPU foam. However, according to the results of AFM, the main difference between MMTNS-1 and MMTNS-2 is thickness. Because the thickness of MMTNS-1 is about 10 times than that of MMTNS-2, while the lateral dimension of MMTNS-1 is only twice than that of MMTNS-2. Therefore, we think the thickness of MMTNS is critical to suppress the flammability of FPU foam. For the effect of lateral dimension of MMTNS on the flammability of FPU foam, we will investigate it in the next work.

(2) Please provide the reason that electrostatic interaction between MMTNS-2 particles is much bigger than the electrostatic interaction between MMTNS-1 particles.

Response: According to the cited reference, the conclusion can be inferred from the calculation formula of electric double layer repulsive potential energy:

UR=64πn0RKTγ02exp(-κH)/κ2

UR∝γ02

γ0=[exp(zeΨ0/2kT)-1]/[exp(zeΨ0/2kT)+1]

In this formula, for MMTNS-1 and MMTNS-2, the only different parameter is Ѱ0, and Ѱ0 is the surface potential of MMT and Ѱ0 ≈ ζ. The γ0 of MMTNS-2 is bigger than MMTNS-1, due to the ζ value of MMTNS-2 (-42mV) is much bigger than that of MMTNS-1(-25 mV). Therefore, the electrostatic interaction between MMTNS-2 particles is much bigger than the electrostatic interaction between MMTNS-1 particles.

(3) In p8, "However, the result is unexpected, because the flammability of MMTNS-2 coated foam with a lower addition is higher than that of MMTNS-1 coated foam." This sentence can be wrong.

Response: Thank you for your reminder, and this sentence has been corrected in the manuscript.

Round 2

Reviewer 2 Report

The revision is O.K. This paper can be accepted in present form.